# BUSClean: Open-source software for breast ultrasound image pre-processing and knowledge extraction for medical AI

**Arianna Bunnell**[1,2]*, **Kailee Hung**[1], **John A. Shepherd**[2], **Peter Sadowski**[1]

**1** Department of Information and Computer Sciences, University of Hawai'i at Mānoa, Honolulu, HI, United States of America, **2** University of Hawai'i Cancer Center, Honolulu, HI, United States of America

* abunnell@hawaii.edu

**Data Availability Statement:** Public sharing of data from the HIPIMR is not explicitly allowed in the protocol approved by our ethics board. Release of data (clinical breast ultrasound images) risks

## Abstract

Development of artificial intelligence (AI) for medical imaging demands curation and cleaning of large-scale clinical datasets comprising hundreds of thousands of images. Some modalities, such as mammography, contain highly standardized imaging. In contrast, breast ultrasound imaging (BUS) can contain many irregularities not indicated by scan metadata, such as enhanced scan modes, sonographer annotations, or additional views. We present an open-source software solution for automatically processing clinical BUS datasets. The algorithm performs BUS scan filtering (flagging of invalid and non-B-mode scans), cleaning (dual-view scan detection, scan area cropping, and caliper detection), and knowledge extraction (BI-RADS Labeling and Measurement fields) from sonographer annotations. Its modular design enables users to adapt it to new settings. Experiments on an internal testing dataset of 430 clinical BUS images achieve >95% sensitivity and >98% specificity in detecting every type of text annotation, >98% sensitivity and specificity in detecting scans with blood flow highlighting, alternative scan modes, or invalid scans. A case study on a completely external, public dataset of BUS scans found that BUSClean identified text annotations and scans with blood flow highlighting with 88.6% and 90.9% sensitivity and 98.3% and 99.9% specificity, respectively. Adaptation of the lesion caliper detection method to account for a type of caliper specific to the case study demonstrates the intended use of BUSClean in new data distributions and improved performance in lesion caliper detection from 43.3% and 93.3% out-of-the-box to 92.1% and 92.3% sensitivity and specificity, respectively. Source code, example notebooks, and sample data are available at https://github.com/hawaii-ai/bus-cleaning.

## Introduction

The development of artificial intelligence (AI) for diagnosis and treatment planning from medical imaging is an exploding area of research. Medical imaging AI being developed for breast ultrasound imaging (BUS) is no exception, with 103 papers using breast, ultrasound and AI in the title/abstract indexed by PubMed since 2023 alone. Data curation and cleanliness

compromising patient privacy. Data may be requested for research purposes at any time through an online research data use request (https://hipimr.shepherdresearchlab.org/data-use-request/).

**Funding:** K. H. was supported by the National Science Foundation Award No. 2149133 while completing this work. https://www.nsf.gov/ The funder did not play any role in the study design, data collection and analysis, decision to publish, or preparation of the manuscript. J.A.S. NCI Grant 5R01CA263491-02. National Cancer Institute. https://www.cancer.gov/ The funder did not play any role in the study design, data collection and analysis, decision to publish, or preparation of the manuscript.

**Competing interests:** The authors have declared that no competing interests exist.

are closely linked to the quality and robustness of developed AI systems. Curation of clinical medical imaging data presents a significant challenge for researchers. In contrast to highly-standardized modalities such as screening mammography, wherein standardized views with little operator-dependent annotation are captured, clinical BUS data may contain a variety of burnt-in annotations, artifacts, and other scan abnormalities. For example, a single diagnostic BUS image may include all the following artifacts burnt-into the image: calipers for lesion measurement, free-text exam positioning information, free-text notes on patient symptoms, blood flow highlighting, and overlays describing software settings. If not removed, cleaned, or otherwise accounted for, these abnormalities introduce noise into the medical imaging AI development pipeline and may result in unexpected relationships between abnormalities being learned by the model or artificial inflation of model performance [1,2]. [3] presents an example of the positive effect of data cleaning when developing AI for BUS specifically. Deep learning model training and evaluation procedures rely on large quantities of relatively clean data to be available, necessitating a mostly-automatic curation and cleaning pipeline, otherwise the volume of data to be manually reviewed becomes unmanageable. For example, [4] develops a BUS AI system trained on five million clinical images which were automatically curated through an internal process described in [5].

In this work, we introduce BUSClean: an open-source software for curation of clinical datasets of BUS images for ingestion into AI development or evaluation pipelines. As part of BUSClean we propose methods for the identification of invalid and enhanced BUS scans, recognition, and classification of text annotations into meaningful categories, and automatic cropping to the scan area. The main contributions of this paper are summarized as follows: (1) release of the only open-source software solution for automatic BUS dataset curation; (2) the first application, to the authors' knowledge, of OCR techniques for detection of scan position, anatomy, and procedure from sonographer annotations; (3) demonstration of BUSClean on unseen internal data showing retained high performance; and (4) presentation of a case study showing the intended use of BUSClean for application in specific clinical data distributions.

## Materials and methods

BUSClean was developed to aid in the preparation of clinical BUS imaging datasets for use in deep learning training and evaluation pipelines using the OpenCV [6] and PIL [7] libraries in Python. Fig 1 gives an outline of the flow and functions of the software. Our algorithm consists of a sequence of preprocessing steps. The steps are modular and can be added or removed by the user. All functions of BUSClean can be categorized into either scan *filtering*, wherein we are flagging scans with certain artifacts, scan *cleaning*, wherein we are applying transformations to the scans themselves to increase uniformity, or *knowledge extraction*, wherein we are extracting metadata from the scan image.

Scan *filtering* is performed to remove scans with disruptive artifacts from development datasets for deep learning training and evaluation. Scan *cleaning* removes the artifact itself while retaining the image. Without extensive cleaning, artifacts may lead to artificially diminished or increased AI performance being reported when using opportunistic or clinical datasets. For example, if a sonographer places measurement calipers on a BUS scan to indicate a lesion and this scan is used to train a lesion detection model, the model may incorrectly learn to detect calipers rather than the actual lesions. An AI model trained on uncurated BUS data may also learn to "cheat" in predictions by using features in the image that are unrelated to the task but correlated with the target variable. This phenomenon is known as the "clever Hans effect" in the machine learning literature. [8–10] present examples of the "clever Hans effect" being discovered in medical AI. As an example, sonographers may be more likely to include

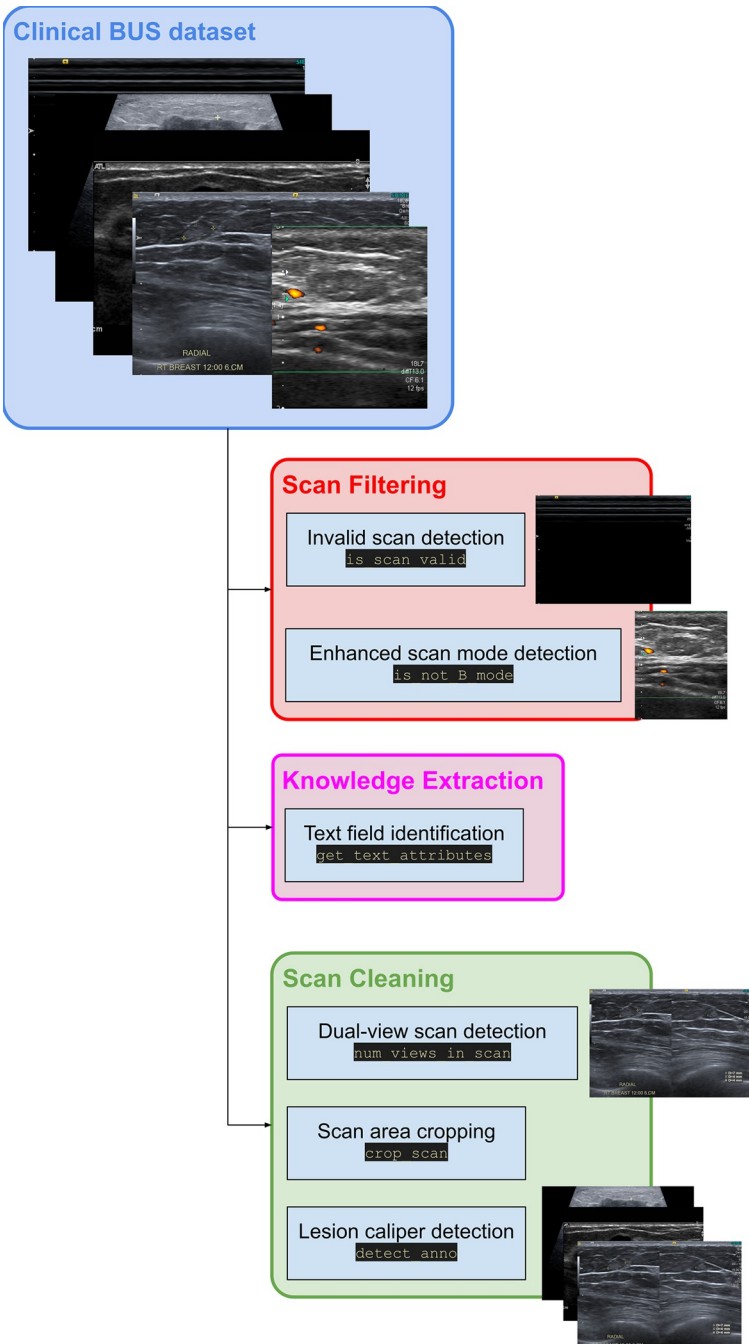

**Fig 1. BUSClean process diagram.** Complete process diagram depicting all functions of the BUSClean open-source software for the purpose of standardizing and extracting metadata from scans before AI model training or evaluation.

text annotations onto a scan which has a higher likelihood of malignancy, due to an increase in notable features or patient symptoms. An AI model then trained on this data may learn to falsely associate the presence of text annotations with lesion malignancy, rather than identifying features of the breast tissue or lesion as being indicative of malignancy.

BUS scans are typically stored as DICOM (Digital Imaging and Communications in Medicine) images, containing embedded header fields which store metadata about the exam. We

found that in our internal clinical dataset, the scan features and artifacts that BUSClean detects could not be identified by examination of the DICOM header alone.

## Scan filtering

BUS scans can be captured in unenhanced brightness mode (B-mode), with blood flow highlighting (Color or Power Doppler highlighting), or with elastography (US modality which provides information about tissue stiffness). In medical AI training, it is desirable to define your population of images precisely (such as only shear-wave elastography BUS images) to reduce the amount of non-task-related variance present in your dataset which may add noise to your model learning. Typically, in deep learning model development for BUS, researchers implicitly define their population of images as B-mode images. [4,11–13] are some examples of models developed for B-mode imaging. There is also a body of work which develops AI for BUS elastography imaging [14–17]. In medical AI evaluation, presentation of images unfamiliar to a model, for example a model trained on Color Doppler images being presented with B-mode images, may make the model perform unexpectedly. To reduce the risk of images outside of the target population being included in AI model pipelines, we develop two scan *filtering* methods to enable the flagging and optional removal of elastography, blood flow highlighted, and invalid BUS images from clinical BUS datasets.

Scans are defined as invalid if more than 75% of the scan area is black (grayscale pixel value of less than five). This method identifies scans which contain very little tissue, scans which only contain burnt-in annotations (such as chaperone name), and scans where the BUS machine malfunctioned. This method may also identify and flag scans in which most of the scan area is a hypoechoic lesion or implant.

Elastography and scans with blood flow highlighting are considered broadly to be "non-B-mode" scans in BUSClean. These scans are identified through a four-step process. First, grayscale scans are removed from the process and labeled as B-mode. Second, HSV color masks for colors found in blood flow highlighting (orange, green, yellow, red, and blue depending on scan manufacturer and mode) and elastography/blood flow highlighting indicator boxes (green and white) are created and dilated. Third, complete and partial rectangles, as well as image-spanning lines are identified from the green/white threshold image and if present, the image is classified as non-B-mode. Lastly, remaining scans are labeled as non-B-mode if the blood flow highlighting mask comprises more than 0.5% of the scan area. The complete process is displayed for three example scans in Fig 2.

## Knowledge extraction

Sonographers will often annotate BUS scan images with text annotations burnt-into the image file to provide additional context for image interpretation (i.e., indicating an area of pain or a palpable lump), anatomical information (i.e., labeling the nipple location or an area of scarring), and descriptive information about the scan collection process. The American College of Radiology's (ACR) Breast Imaging Reporting & Data System (BI-RADS) system for US outlines guidelines in Section IC: Labeling and Measurement that all BUS exams should contain permanent identification containing the following: (1) facility name and location; (2) examination date; (3) patient's first and last name; (4) patient identification number or date of birth; (5) designation of left or right breast; (6) Anatomic location using clock-face notation or a labeled diagram; (7) Transducer orientation; (8) distance from the nipple in centimeters; and (9) sonographer's and/or physician's identifying number/initials/symbol [18].

Items (1)–(4) and (9) from the ACR labelling guidelines for BUS constitute personally identifying information which is typically stored in the DICOM header metadata and removed

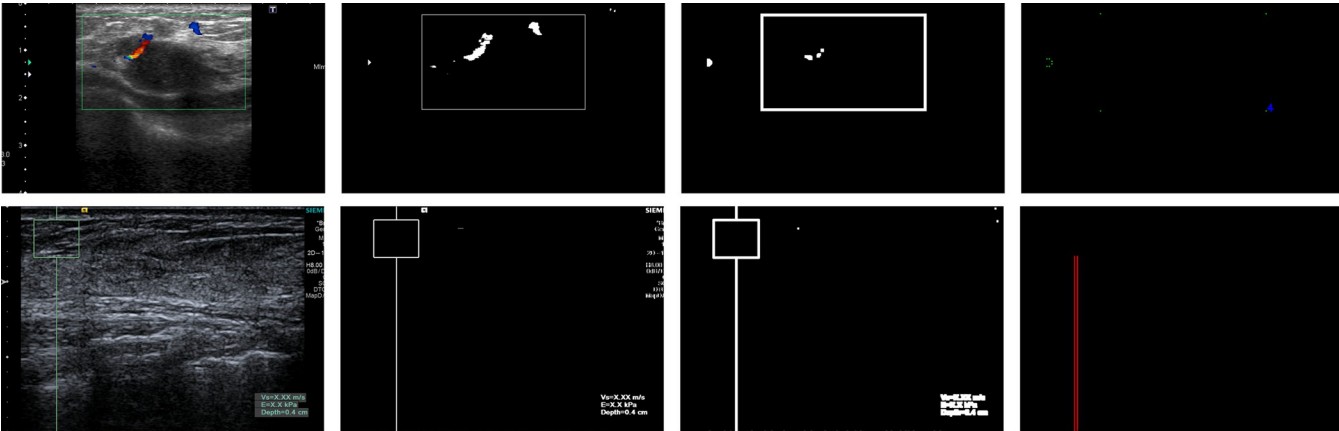

**Fig 2. Detecting non-B-mode scans.** Diagram showing the step-by-step processing of detecting non-B-mode scans. The top row shows a BUS scan with blood flow highlighting being successfully detected with four contours detected, indicating a rectangle in the image which is characteristic of blood flow highlighting being applied. The bottom row shows an elastography scan being successfully detected via recognition of the extended vertical line in the elastography overlay.

prior to scans being exported from clinical centers for deep learning training. Items (5)–(8), however, contain non-identifiable information about the scan itself, are usually burnt-in to the image, and may contain information useful for AI training or evaluation. For example, extraction of laterality and distance from nipple or abnormality may allow for automatic filtering of scans which are unlikely to include a lesion prior to classification or segmentation, AI training, or consulting radiologist labeling.

BUSClean identifies items (5)–(8) from BUS scan images using a combination of Optical Character Recognition (OCR) through EasyOCR (Jaided AI; Bangkok, Thailand) and regular expression-based pattern matching. Notably, BUSClean does not recognize quadrant-style notation (LOQ for lower outer quadrant, RIQ for radial inner quadrant, etc.) for anatomic location, as this style is not explicitly recommended by the ACR in their guidelines [18]. BUSClean only automatically recognizes anatomical location in clock-face notation, which is the text method recommended by the ACR [18]. See Fig 3 for an illustration of BUSClean text field matching.

In addition to the ACR-defined items, BUSClean also identifies and flags scans with text indicating what we define as *procedural* imaging, lesion measurements, and imaging of the axillary region. *Procedural* imaging, according to our definition, is any imaging containing text which refers to a surgical procedure, such as US-guided biopsy. We also include imaging which notes inclusion of a clip, marker, or coil into this category. Due to the abnormality of these findings, they may be worth excluding in deep learning model development pipelines. AI models trained to recognize lesions as tissue abnormalities may erroneously flag clips, markers, or coils as lesions due to high degree of contrast with surrounding tissue. Lesion measurements may or may not coincide with measurement calipers (see Fig 4 for an example) and could be used for automatically excluding lesions smaller or larger than the target population, such as in [20,21]. Imaging of the axillary region is typically focused on the lymph nodes for detection of cancer metastasis, rather than imaging of a specific lesion or area of interest, making them outside the target image population for AI models designed for lesion detection and segmentation from BUS.

## Artifact detection

After capture of a scan frame showing a lesion or mass of interest, the examining sonographer may choose to annotate the frame with measurement calipers or capture another orthogonal

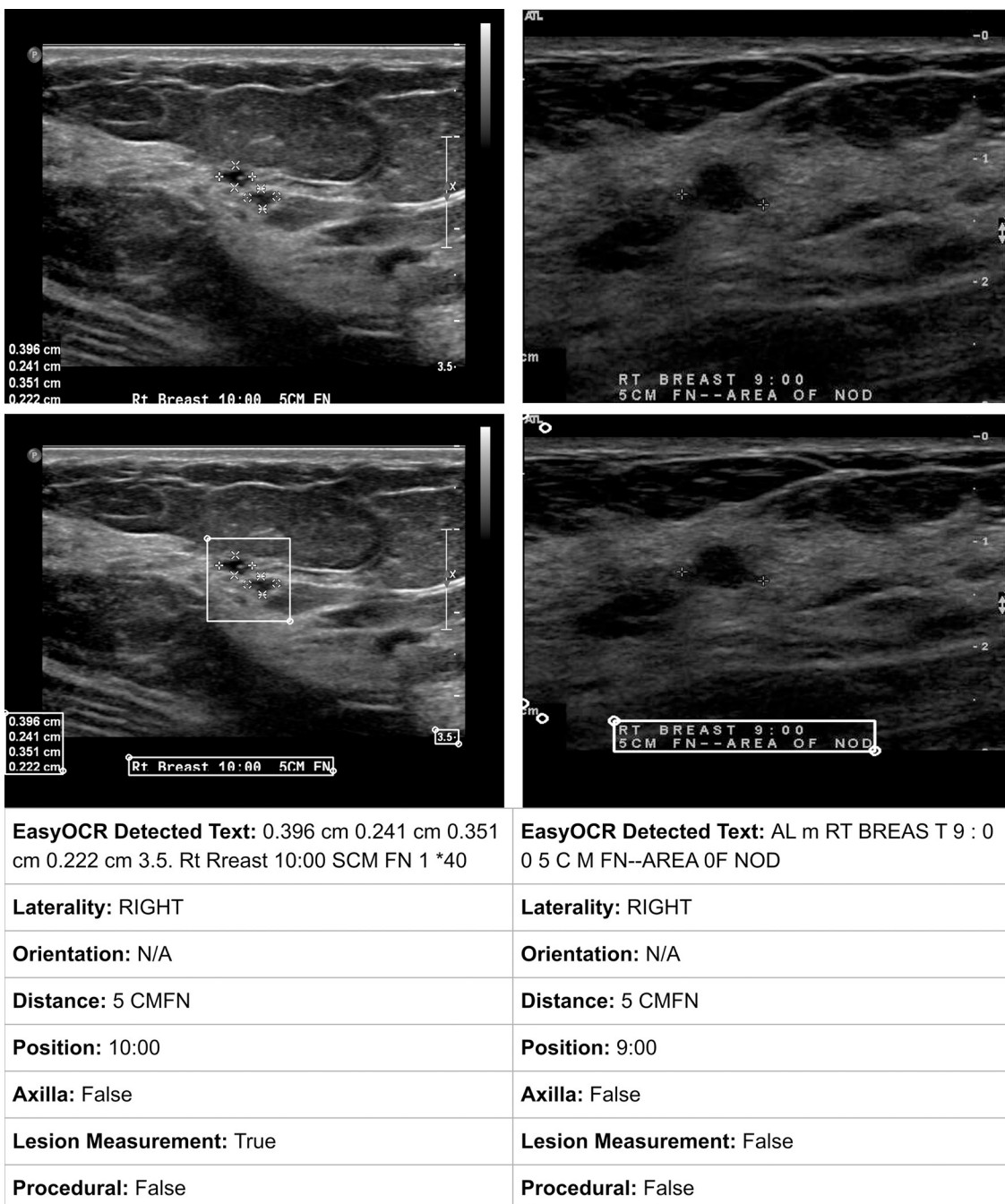

**Fig 3. BUSClean text field matching.** Two example scans with text detected by EasyOCR [19] and the fields recognized after pattern matching by BUSClean.

view to be displayed alongside the initial frame in a single imaging record. The presence of lesion calipers in BUS scans used for AI model development may lead to hard-to-detect biases in model decision-making which are not frequently mentioned in BUS AI model development. Many papers which develop AI on the BUSI dataset [22] in particular do not mention manual or automatic filtering of lesion calipers prior to the training or evaluation of their methods; some examples are [23–26]. Failure to remove lesion calipers is problematic for

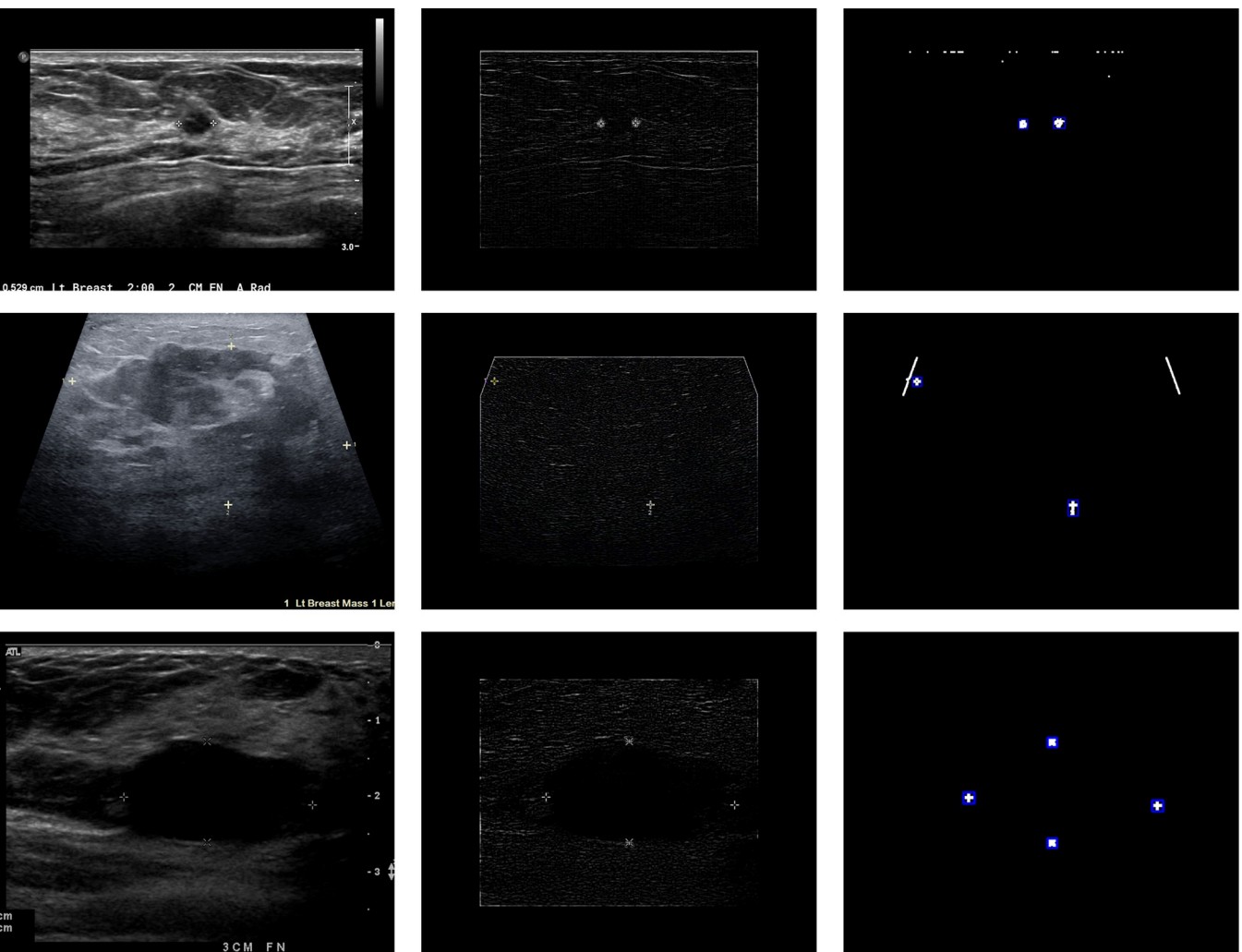

**Fig 4. Lesion caliper detection process diagram.** Diagram showing the step-by-step process of detecting calipers in a BUS scan. The top and bottom rows show scans where all calipers are successfully detected and the image flagged for caliper presence. The middle row shows a scan where two of the four calipers are initially cropped out, but the scan is still identified as having calipers because two remain in the reduced scan area.

methods performing lesion segmentation or detection, as the location of the lesion is explicitly indicated by calipers, leaving the AI to only interpolate lesion boundaries between sonographer-placed calipers, possibly completely failing to learn how to locate lesions without these markings. As it is implemented in BUSClean, lesion caliper identification may fall into either the scan *filtering* or *cleaning* subset of functions.

Dual-view scans (a single DICOM image which contains two images displayed side-by-side for purposes of comparison between two modalities or perspectives of a lesion; see an example in Fig 1) contain a harsh dividing line in the middle of the image which may confuse AI systems in evaluation and do not represent any properties of breast tissue. Leaving dual-view scans in evaluation datasets may lead to artificially low performance results being reported. Additionally, dual-view scans may contain views of two different lesions, one cancerous and one benign, confusing the histological labeling of BUS scans. Dual-view scan identification falls into the scan *cleaning* subset of BUSClean functions.

Dual-view scans are identified though the following process: (1) teal/green color masks are used to filter out elastography scans; (2) scan height/width ratio is used to filter out scans with

width < 75% of their height; (3) application of a Canny edge detection filter; (4) split detection through calculation of number of edge-pixels on the midline is both > 100 and greater than 10 + number of white pixels at (midline– 10 pixels), and 10 + number of white pixels at (midline + 10 pixels).

Lesion calipers can be white/gray or colored and shaped as crosses, numbers, or "X"s and may or may not be connected with dotted lines spanning the lesion. In BUSClean, scans with lesion calipers are flagged and the coordinates of the detected calipers returned for optional user-side cropping. Lesion calipers are detected via a two-step process. First, the scan itself is enhanced by black masking of the outside 15% along every dimension to reduce the likelihood of software overlays or text being mistakenly identified as calipers. Then, the FIND-EDGES and maximum filters from the PIL [7] library are applied to isolate the caliper shapes and denoise the edge image. Finally, the resulting image is dilated. For the second step, contours are detected in this enhanced image and if the contour's bounding box is between 70 and 10 pixels in height and width, it is counted as a caliper and the bounding box coordinates returned. Fig 4 displays the two-step processing on three example BUS images.

## Scan area cropping

Identification and cropping of the scan area prior to AI training removes irrelevant background pixels and software artifacts from the scan and increases the effectiveness of standard image data augmentation methods such as rotation by making the resulting augmented images more plausible. BUSClean's cropping method is designed to preserve as much tissue in the scan area as possible, while minimizing the amount of background in the final cropped image. Our two-stage method for scan area identification and cropping is adapted from the shape-based cropping method presented in [5]. Scan area cropping falls into the scan *cleaning* subset of BUSClean functions.

Initially, background pixels are identified and cropped by using binary thresholding (pixel values > mode pixel value + 10), erosion and dilation, and isolation of the largest connected component in the resulting mask. The bounding box surrounding the largest connected component is the first set of cropping coordinates. For rectangular scans, the process ends here, as the scan area completely fills the bounding box.

In addition to rectangular scans, BUS scan areas may also be convex, trapezoidal, or irregular (due to excessive shadowing in the lower half of the scan). These scan types benefit from a more aggressive cropping strategy. We define a Euclidean coordinate system on our BUS scan image where in the topmost, leftmost pixel corresponds to (0,0) and the bounding box is defined by $(x_{left}, y_{top}, x_{right}, y_{bottom})$ with width $w$ and height $h$. The width of the scan is cropped to the median index of the leftmost (lowest) and rightmost (highest) non-mode-valued pixel in the following image slices: $\left[y_{top}, {}^h/_3 + y_{top}\right)$, $\left[{}^h/_3 + y_{top}, {}^{2h}/_3 + y_{top}\right)$, and $\left[{}^{2h}/_3 + y_{top}, y_{bottom}\right]$. The height of the scan is cropped according to a similar strategy to the median index of the topmost (lowest) and bottommost (highest) non-mode-valued pixel in the following image slices: $\left[x_{left}, {}^w/_3 + x_{left}\right)$, $\left[{}^w/_3 + x_{left}, {}^{2w}/_3 + x_{left}\right)$, and $\left[{}^{2w}/_3 + x_{left}, x_{right}\right]$. Fig 5 displays the described coordinate system over simulated binary images with convex, trapezoidal, and irregular scan area shapes.

## Development dataset

BUSClean was developed on an internal dataset of 2,000 BUS scans from the Hawai'i and Pacific Islands Mammography Registry (HIPIMR; Western-Copernicus Group IRB Study Number 1264170). Researchers did not have access to information that could identify

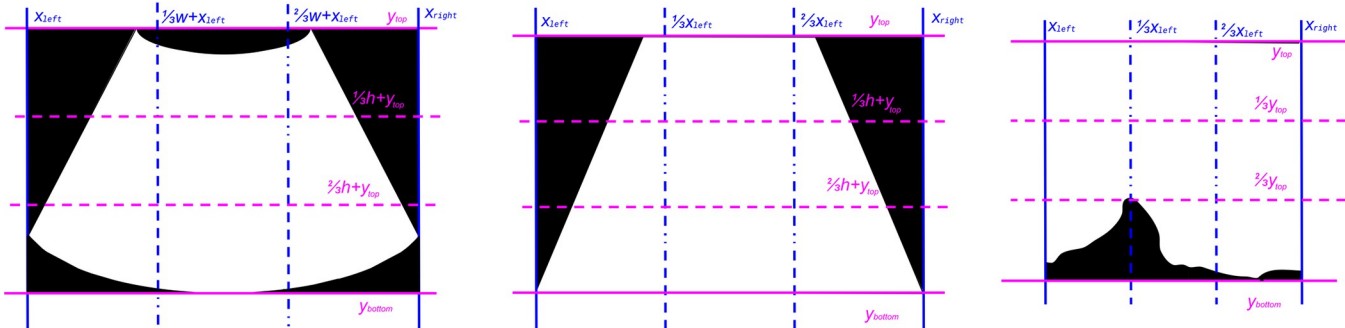

**Fig 5. Scan cropping coordinate system.** Diagram displaying the coordinate system overlaid on scans with convex (left), trapezoidal (middle), and irregular (right) scan areas.

individual participants during or after data collection. The requirement for informed consent was waived. HIPIMR data are available for research via an online data use request. All images were collected from 2009–2023 on the island of Oʻahu. The HIPIMR collects data from three clinical partners: Clinic 1 is a nonprofit healthcare network (comprising four medical centers); Clinic 2 is a private nonprofit tertiary hospital; and Clinic 3 is a diagnostic medical imaging center. BUS images were selected for inclusion via a simple random sample from all images from all women in the HIPIMR with negative, benign, or probably benign (BI-RADS 1, 2, or 3) BUS visit within one year of a negative screening mammography visit and no personal history of breast cancer. Images from 1,413 patients were included. In the development set, 81% of BUS images were captured on a Philips Medical Systems IU22 system; 11.85% on a Philips ATL HDI 5000; 6.2% on a Toshiba Aplio XG; 0.85% on a Siemens ACUSON S2000; and 0.1% on an Esaote Technos 8234. Data were accessed for research purposes on February 2, 2023. For illustrative purposes, Fig 6 provides examples of all scan and text types present in the development dataset which BUSClean was designed to automatically identify.

### Internal test dataset

We designated an internal dataset of BUS images to test the performance of BUSClean's knowledge extraction and artifact detection on unseen data. The internal test dataset is composed of 430 unseen BUS images. BUS images were selected for inclusion in the internal testing dataset via a simple random sample from all images (excluding development set) from all women in the HIPIMR with negative, benign, or probably benign (BI-RADS 1, 2, or 3) BUS visit within one year of a negative screening mammography visit and no personal history of breast cancer. Images from 391 patients were selected for inclusion, where 174 patients have images included in both the development and internal testing sets.

### Case study dataset

We also examine the performance of BUSClean on an external dataset to perform a case study of how the tool may be used by deep learning researchers in the field as well as further testing performance of BUSClean's knowledge extraction and artifact detection on unseen data. The external test dataset for the case study is comprised of the 780-image BUSI dataset from [22]. This dataset of BUS images is available publicly and was collected in 2018 from Baheya Hospital for Early Detection and Treatment of Women's Cancer, Cairo (Egypt) with LOGIQ E9 ultrasound system and LOGIQ E9 Agile ultrasound. In contrast to the internal test and development datasets, the external test dataset was collected experimentally, rather than

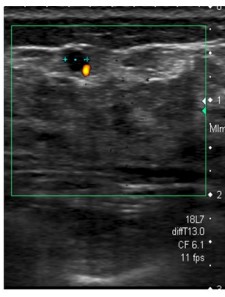
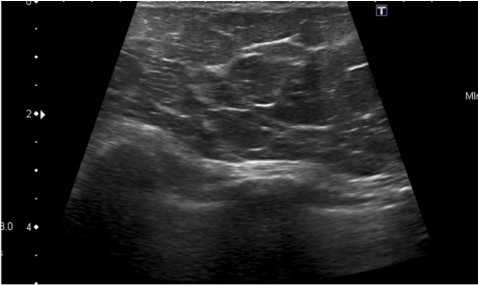
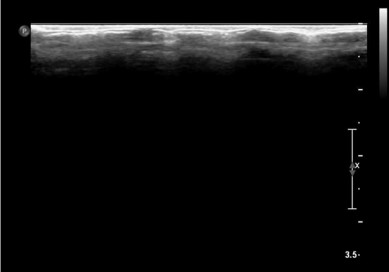

Blood flow highlighted scan (enhanced scan mode) with lesion calipers

B-mode scan with trapezoidal scan area (specialized cropping procedure)

Invalid scan

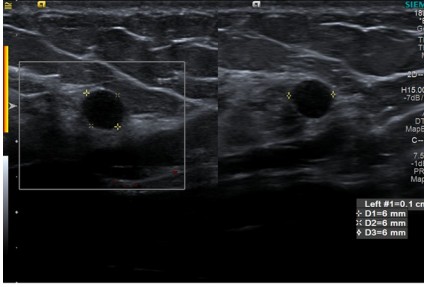
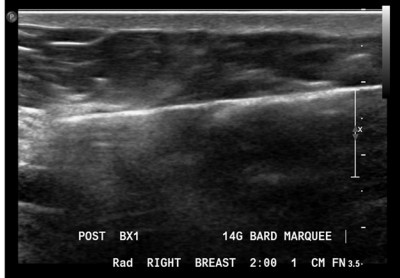
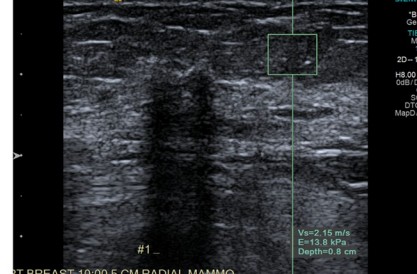

Dual-view, blood-flow highlighted scan (enhanced scan mode) with lesion calipers and lesion measurement text annotations

B-mode scan with procedural, laterality, orientation, distance, position, and lesion measurement text annotations.

Elastography scan (enhanced scan mode) with laterality, distance, and position text annotations.

**Fig 6. Development dataset example images.** Six example images from the development dataset of BUSClean showing all types of scans flagged through scan filtering (enhanced scan mode and invalid scans), scan cleaning (lesion calipers, dual-view scans, and rectangular/trapezoidal scans) and knowledge extraction methods (all text annotation types including procedural imaging and lesion measurements).

opportunistically from clinical care. Table 1 displays population counts for all tested characteristics on the BUSI dataset.

The BUSI dataset originates from outside the U.S. (Egypt), and therefore is not expected to follow the ACR BUS labelling guidelines. Thus, the case study was tested for text recognition only, not classification into the ACR BUS labelling characteristics. Additionally, the scan filtering and artifact detection (specifically calipers) were tested on the case study dataset. Due to the high level of curation of the BUSI dataset, it can be presumed there are no invalid scans and the invalid scan filtering method was not tested on it. BUSClean's functions are modular and can be selectively applied by researchers according to the characteristics of their BUS data.

**Table 1. Sample characteristics for the internal test and case study (BUSI) [22] datasets.**

| Characteristic, N (%) | Test Dataset | |
|---|---|---|
| | **Internal** | **Case Study** |
| Images | 430 | 780 |
| Images with any artifact type | 411 (95.6) | 215 (27.6) |
| Images with non-text artifact | 101 (23.5) | 134 (17.2) |
| Images with calipers | 89 (20.7) | 127 (16.3) |
| Invalid images | 2 (0.47) | - |
| Dual-view images | 4 (0.93) | - |
| Images with text annotations | 398 (92.6) | 132 (16.9) |
| Procedural images | 31 (7.21) | - |
| Images with blood flow highlighting | 10 (2.32) | 11 (1.41) |
| Elastography images | 0 (0.00) | 0 (0.00) |

## Statistical analysis

The ground truth for all tasks except scan area cropping was determined through human annotation. The development, case study, and internal testing datasets were labeled for scan abnormalities and modality by two authors (A.B. and K.H.) with disagreements in labeling resolved through adjudication. Performance was assessed via computation of sensitivity and specificity for each individual binary classification task (i.e., labeling a scan as invalid/not invalid).

## Results

### Internal test dataset

In the internal test set, 82.33% of BUS images were captured on a Philips Medical Systems IU22 system; 10.47% on a Philips ATL HDI 5000; 5.6% on a Toshiba Aplio XG; and 1.6% on a Siemens ACUSON S2000.

The scan filtering, knowledge extraction, and artifact detection pipelines were tested on the internal test dataset. Table 1 displays population counts for all tested characteristics on the internal dataset.

### Performance

Confusion matrices and statistics of BUSClean performance on the internal test dataset are presented in Table 2. BUSClean identified the laterality, orientation, distance, position, axilla, lesion measurement, and procedural text properties with 99.2%, 97.6%, 96.7%, 100%, 95.8%, 97.5%, and 100% sensitivity and 98.2%, 100%, 98.7%, 100%, 100%, 98.3%, and 100% specificity, respectively (for each property) on the internal test dataset. Scans imaged with an enhanced scan mode (non-B-mode; elastography scans or scans with blood flow highlighting) were identified with 100% sensitivity and 99.5% specificity in the internal test dataset. Lesion calipers, dual-view scans, and invalid scans were detected with 96.7%, 100%, and 100% sensitivity and 93.3%, 98.6%, and 100% specificity in the internal dataset.

**Table 2. Confusion matrix and performance statistics of BUSClean performance on the internal test dataset.**

| | TP[a] | FP | TN | FN | Sens. | Spec. | F1 |
|---|---|---|---|---|---|---|---|
| **Text annotation recognition** | | | | | | | |
| Laterality | 371 | 1 | 55 | 3 | 0.992 | 0.982 | 0.995 |
| Orientation | 289 | 0 | 133 | 7 | 0.976 | 1.000 | 0.986 |
| Distance | 264 | 2 | 153 | 9 | 0.967 | 0.987 | 0.980 |
| Position | 54 | 0 | 376 | 0 | 1.000 | 1.000 | 1.000 |
| Axilla | 159 | 0 | 264 | 7 | 0.958 | 1.000 | 0.979 |
| Lesion measurement | 78 | 6 | 344 | 2 | 0.975 | 0.983 | 0.951 |
| Procedural | 31 | 0 | 399 | 0 | 1.000 | 1.000 | 1.000 |
| **Other scan abnormality** | | | | | | | |
| Enhanced scan mode | 10 | 2 | 418 | 0 | 1.000 | 0.995 | 0.909 |
| Invalid scan | 2 | 0 | 428 | 0 | 1.000 | 1.000 | 1.000 |
| Dual-view scan | 4 | 6 | 420 | 0 | 1.000 | 0.986 | 0.571 |
| Caliper presence | 86 | 23 | 318 | 3 | 0.967 | 0.933 | 0.869 |

TP = true positive; FP = false positive; TN = true negative; FN = false negative; Sens = sensitivity; Spec = specificity; F1 = F1 score.

[a]Detections of a text property are only considered to be TP if the field is both recognized and classified correctly.

**Table 3. Confusion matrix and performance statistics of BUSClean performance on the case study dataset/BUSI dataset [22].**

|  | TP | FP | TN | FN | Sens. | Spec. | F1 |
|---|---|---|---|---|---|---|---|
| Text presence | 117 | 11 | 637 | 15 | 0.886 | 0.983 | 0.900 |
| Enhanced scan mode | 10 | 1 | 768 | 1 | 0.909 | 0.999 | 0.909 |
| Caliper presence | 55 | 44 | 609 | 72 | 0.433 | 0.933 | 0.487 |

TP = true positive; FP = false positive; TN = true negative; FN = false negative; Sens = sensitivity; Spec = specificity; F1 = F1 score.

Confusion matrices and statistics of BUSClean performance on the case study/BUSI dataset are presented in Table 3. BUSClean recognized the presence of text with 88.6% sensitivity and 98.3% specificity in the case study dataset. Scans imaged with an enhanced scan mode were identified with 90.9% sensitivity and 99.9% specificity in the case study dataset. Lesion calipers were detected with 43.3% sensitivity and 93.3% specificity in the case study dataset using out-of-the-box BUSClean.

## Discussion

### Case study

The performance of BUSClean on the internal test dataset of unseen images was high. However, there will always be new types of annotations to handle, so researchers should anticipate needing to customize the functions BUSClean provides to their needs. BUSClean's modular design and open-source license facilitate this. We recommend researchers first apply BUSClean to a small test dataset to evaluate performance on data from any new source. Custom filtering, cleaning, or knowledge extraction methods can be easily added to the processing pipeline. Users are encouraged to share these methods with the community. In the BUSI dataset case study, sensitivity in lesion caliper detection using the out-of-the-box BUSClean method decreased to under 50% (43.3%). Visual inspection of false negative results revealed that all but two false negatives failed to be detected as lesion calipers due to the style of spanning dotted line used to connect markers in the external test dataset (the remaining two were due to tissue brightness masking the caliper). This caused the internal line and markers to be detected as one object after dilation and therefore be missed by BUSClean. See Fig 7 for example false negative scans. This style of annotation was not present in the development dataset and thus was not accounted for in the original BUSClean code.

We implemented a new lesion caliper detection method for the BUSI dataset which additionally performed line detection using the Hough Transform through OpenCV [6] and if at least two detected lines intersected (but were not parallel), a caliper was detected. This change improved the sensitivity to 92.1% (TP = 117 and FN = 10) while only slightly decreasing specificity to 92.3% (TN = 603 and FP = 50). This case study provides an illustration of how BUSClean should be implemented, tested, and customized for use on new research datasets.

### Limitations

The main limitation of BUSClean is that there is no guarantee BUSClean's methods will transfer to new clinical BUS data-generating distributions, but the open-source nature of the software enables allow developers to easily adapt these methods future datasets. Despite our dataset size and diversity, researchers may encounter new types of lesion caliper or scan abnormalities in their datasets and should monitor BUSClean's classifications to ensure adequate performance.

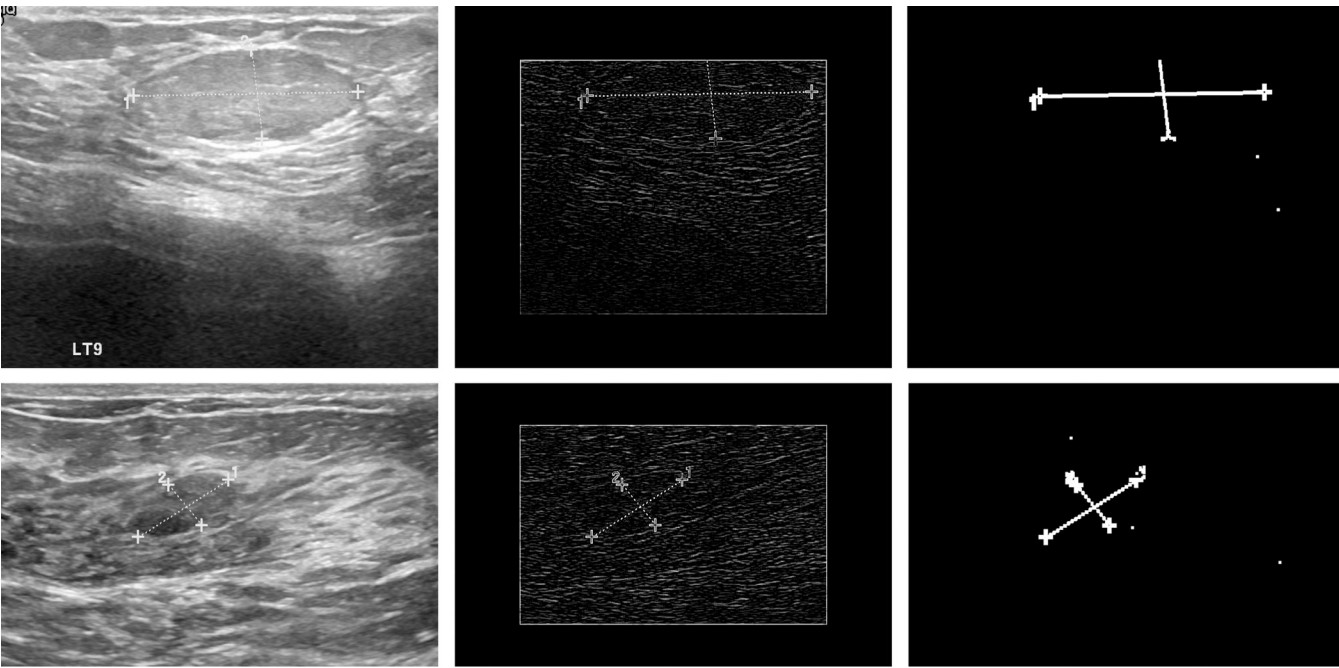

**Fig 7. False-negative caliper detections in case study.** Example false negative (FN) scans from the external test (BUSI) [22] on the lesion caliper detection task. The resulting binary shapes (left) are too large to be detected by BUSClean's caliper detection method.

A second limitation of the BUSClean software is the lack of a built-in cropping method for removing lesion calipers from scans. It is plausible that BUS scans may have lesion calipers on the edge of the scan area, leaving a significant amount of uninterrupted tissue which could be fed to BUS AI pipelines without transferring possibly harmful correlations to model learning. However, we found in our internal dataset that this was not a common occurrence; lesions are frequently the only structure of interest in the scan and likely to be centered by the sonographer for ease of interpretation, leaving little uncorrupted tissue to be cropped and used for model training.

The last limitation of BUSClean is the focus on ACR-defined BUS text classification fields and English text. This limitation can be addressed by individual developers as BUSClean can easily be extended to new annotations and languages by defining new regular expressions and using a different OCR model.

## Conclusions

In this work we presented BUSClean, an open-source software solution for curation of clinical BUS datasets for AI model training and evaluation. We define a pre-processing algorithm consisting of scan *filtering*, scan *cleaning*, and *knowledge extraction* methods. The system was evaluated on two different held out datasets for performance in identifying text fields and BUS scan types. We also demonstrated a case study showing the intended use of BUSClean on new clinical BUS datasets. BUSClean will help researchers curate the image datasets necessary for developing robust AI systems.

## Acknowledgments

The authors would like to acknowledge Dr. Thomas K. Wolfgruber for preparation of the HIPIMR BUS data and the DICOM image extraction process.

## Author Contributions

**Conceptualization:** Arianna Bunnell.

**Data curation:** Arianna Bunnell.

**Formal analysis:** Arianna Bunnell.

**Funding acquisition:** John A. Shepherd.

**Investigation:** Arianna Bunnell, Kailee Hung.

**Methodology:** Arianna Bunnell, Kailee Hung, Peter Sadowski.

**Project administration:** Arianna Bunnell.

**Resources:** John A. Shepherd.

**Software:** Arianna Bunnell, Kailee Hung.

**Supervision:** John A. Shepherd, Peter Sadowski.

**Validation:** Kailee Hung.

**Writing – original draft:** Arianna Bunnell.

**Writing – review & editing:** John A. Shepherd, Peter Sadowski.

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
