## [Decision Letter · Decision Letter 0]

1 Oct 2024

PONE-D-24-32854BUSClean: Open-source software for breast ultrasound image pre-processing and knowledge extraction for medical AIPLOS ONE

Dear Dr. Bunnell,

Thank you for submitting your manuscript to PLOS ONE. After careful consideration, we feel that it has merit but does not fully meet PLOS ONE’s publication criteria as it currently stands. Therefore, we invite you to submit a revised version of the manuscript that addresses the points raised during the review process.

We look forward to receiving your revised manuscript.

Kind regards,

Arka Bhowmik, Ph.D.

Academic Editor

PLOS ONE

**Journal Requirements:**

K. H. was supported by the National Science Foundation Award No. 2149133 while completing this work. https://www.nsf.gov/ The funder did not play any role in the study design, data collection and analysis, decision to publish, or preparation of the manuscript.

J.A.S. NCI Grant 5R01CA263491-02. National Cancer Institute. https://www.cancer.gov/ The funder did not play any role in the study design, data collection and analysis, decision to publish, or preparation of the manuscript.

**Additional Editor Comments:**

This paper presents a breast ultrasound image text identification, detail extraction, errors in scan, different image protocols, and text annotation removal application, specifically used for downstream AI pipeline. My recommendations for enhancing the paper are listed below

1. Abstract: The abstract could identify the unique purpose of this open-source tool. It should also explicitly specify the tasks this tool could do. For instance, instead of simply cleaning specify the cleaning operations. Similarly, instead of detail extraction specify details like laterality, lesion size, etc from images instead of dicom header.

2. Introduction: Although some points were discussed in current introduction, but it lacks critical voice. Therefore, introduction need to be rewritten to add following points as well (a) what are the unavoidable features that are found in BUS which needed filtering or cleaning? (b) importance of pre-processing these unwanted artifacts or irregular scans, for downstream application and downstream clinical procedures, (c) what are the gaps/lacunae in previous implementation, (d) which of these gaps this application is aiming to address. Using this structure, authors could expand the critical need for all readers.

3. Methods:

3.1. Overall, the methodology section and figures can be improved.

3.2. Flow diagram: A flow diagram to display BUS dataset distribution (i.e., number of internal or external) for application development and validation can be included. If available, also include number of exams and no. of unique patients.

3.3. Data collection: The study could detail the essential criteria for patient/BUS selection or exclusion criteria, if any. What un-biased steps were adopted to divide internal data into development and validation dataset (like random sampling of all data/splitting based on manufacturer/all data of a particular center is reserved for test set)?

3.4. Statistical analysis: This section is missing. Evaluation metrics fall under this sub-section. Approach used to determine the ground truth and statistical comparison/significance need to be discussed in this section.

4. Discussions: The study should discuss following limitation of this study.

4.1 The ground truth definition may differ from person to person. In this study, ground truth defined by same two annotators or developers for internal and external dataset. This could explain extraordinarily high performance of a CAD software. This indicates the possible bias in the study design.

Reviewers' comments:

Reviewer's Responses to Questions

**Comments to the Author**

1. Is the manuscript technically sound, and do the data support the conclusions?

Reviewer #1: Partly

2. Has the statistical analysis been performed appropriately and rigorously? 

Reviewer #1: Yes

3. Have the authors made all data underlying the findings in their manuscript fully available?

Reviewer #1: Yes

4. Is the manuscript presented in an intelligible fashion and written in standard English?

Reviewer #1: Yes

5. Review Comments to the Author

**Reviewer #1:** This work developed an open source software, namely BUSClean, for curation of clinical datasets of BUS images for ingestion into AI development or evaluation pipelines. The authors publicly released the software. It is a novel application to detect span position, anatomy, and procedure from sonographer annotations. It demonstrates shows retained high performance. The presentation of a case study shows the intended use for specific clinical data distributions.

1. Some expressions are somewhat subjective and lack objective evidence. It is recommended to cite some references to show that the author's point of view is supported. Especially in the first paragraph of introduction.

2. The importance of the research topic is not sufficient. The author points out that various defects in BUS mislead deep learning models, and gives examples of pre-processing in BUSI systems developed by previous studies, but it is not enough to explain what happens without pre-processing.

3. The authors point out that unlike other imaging data, BUS data contains some defects. However, mere description is very unspecific and difficult to understand. What are the flaws in BUS? What can the software solve? Or all of them? It is suggested that the author include some images for each defect to make it easier for the reader to understand the importance of preprocessing.

4. I'm not absolutely sure, but it seems that Table 2 and Table 3 are not confusion matrices, right?

5. In the performance section, the authors describe the sensitivity and specificity of the method on various identifications. But why not include these results in the table? In addition, why did the author not consider other evaluation indicators such as accuracy, precision, f1 score and AUC for the report? Is such a short result section sufficient to verify the performance superiority of the proposed software?

6. The author's verification of the software developed is not sufficient. The authors only report the results of the software's handling of various defects. However, as this software is a BUS preprocessing software, the author should investigate other evaluation methods for medical image preprocessing. Should authors consider finding out the latest centralized BUS image preprocessing methods and comparing them to show the performance superiority of the software? For example: Article: "KRC-APM: Key region cutting and artificial prior model for breast cancer recognition in ultrasound images" also proposed a BUS preprocessing method.

7. The authors point out that flaws in BUS data can mislead AI models. However, there is no proof in the author's experiment that the resulting images processed by the software can improve the performance of the AI model. Should the authors consider conducting ablation experiments to show that the various modules of the software can reasonably handle different defects, and to verify that the sub-process results and final results of the processing can improve the AI model?

8. The author introduces the processing principle of each module of the software in sufficient detail. However, it is not clear how the different processes of this software deal with different defects. It is suggested that the author give some diagrams for the intermediate and final results of each processing subprocess, so that readers can be more clear about the effectiveness and importance of preprocessing.

9. The author must be more patient and careful to prepare the drawings in the paper. The few figures that do exist look very blurry. In Figure 1, some arrows are covered by the module, while others are in front of the module. The jagged edges of the modules are also very severe. In Figure 3, why is there an extra black rectangle in the bottom half? Do they have any special meaning? The text in Figure 5 is barely legible. The captions in Figure 6 appear to be heavily double shadowing, making them difficult to recognize. The author needs to improve the quality of the graph and suggests using vector graph.

10. The authors' references appear to be slightly out of date and do not indicate the cutting-edge nature of the study. Authors are advised to read and refer to recent literature:

- KRC-APM: Key region cutting and artificial prior model for breast cancer recognition in ultrasound images

- Breast cancer diagnosis using optimized deep convolutional neural network based on transfer learning technique and improved Coati optimization algorithm

- One-step abductive multi-target learning with diverse noisy samples and its application to tumour segmentation for breast cancer

- HoVer-Trans: Anatomy-Aware HoVer-Transformer for ROI-Free Breast Cancer Diagnosis in Ultrasound Images

- REAF: ROI Extraction and Adaptive Fusion for Breast Cancer Diagnosis in Ultrasound Images

6. PLOS authors have the option to publish the peer review history of their article (what does this mean?). If published, this will include your full peer review and any attached files.

Reviewer #1: No

---

## [Author Response · Author response to Decision Letter 0]

28 Oct 2024

We thank the reviewer and the editor for their thoughtful comments on our manuscript. We have carefully considered each of the raised points and prepared point-by-point responses below. We have also moved some sections of the manuscript to accommodate the new Statistical Analysis section requested. 

Response to Additional Editor Comments

Comment 1: The abstract could identify the unique purpose of this open-source tool. It should also explicitly specify the tasks this tool could do. For instance, instead of simply cleaning specify the cleaning operations. Similarly, instead of detail extraction specify details like laterality, lesion size, etc from images instead of dicom header.

Response 1: Thank you for this comment. We have revised the abstract to clearly identify the purpose and the operations performed by BUSClean: “The algorithm performs BUS scan filtering, cleaning, and knowledge extraction from sonographer annotations” has been updated to “The algorithm performs BUS scan filtering (flagging of invalid and non-B-mode scans), cleaning (dual-view scan detection, scan area cropping, and caliper detection), and knowledge extraction (BI-RADS Labeling and Measurement fields) from sonographer annotations.” 

Comment 2: Although some points were discussed in current introduction, but it lacks critical voice. Therefore, introduction need to be rewritten to add following points as well (a) what are the unavoidable features that are found in BUS which needed filtering or cleaning? (b) importance of pre-processing these unwanted artifacts or irregular scans, for downstream application and downstream clinical procedures, (c) what are the gaps/lacunae in previous implementation, (d) which of these gaps this application is aiming to address. Using this structure, authors could expand the critical need for all readers.

Response 2: Thank you for this comment. We have added text which addresses each of the raised points into the introduction. A summary of the changes is included below. 

a) Added the following sentence to the introduction to explain unavoidable features which can be found in clinical BUS imaging: “For example, a single diagnostic BUS image may include all the following artifacts burnt-into the image: calipers for lesion measurement, free-text exam positioning information, free-text notes on patient symptoms, blood flow highlighting, and overlays describing software settings.”

b) We agree additional context may be needed for readers who do not develop medical AI. We have added an example from the literature showing the effect of image cleaning method on BUS AI performance. We have also added three references to the literature on the Clever Hans Effect into the methods section to provide additional background for non-AI expert readers. 

c) There are no existing implementations of cleaning libraries for BUS scans. We have clarified this in the list of contributions for this paper: “The main contributions of this paper are summarized as follows: (1) release of the only open-source software solution for automatic BUS dataset curation…”

Comment 3: A flow diagram to display BUS dataset distribution (i.e., number of internal or external) for application development and validation can be included. If available, also include number of exams and no. of unique patients.

Response 3: Thank you for this comment and we agree. A flow diagram would be useful. However, due to the high number of process-based figures in the paper, and the relatively simple sampling procedure for our image datasets (simple random sample, with no exclusions) we clarified the text for the sampling procedure and added patient counts but did not add another diagram.

Comment 4: The study could detail the essential criteria for patient/BUS selection or exclusion criteria, if any. What un-biased steps were adopted to divide internal data into development and validation datasets (like random sampling of all data/splitting based on manufacturer/all data of a particular center is reserved for test set)?

Response 4: Thank you for this comment. BUS images were selected for inclusion in the internal testing dataset via a simple random sample from all images (excluding the development set) from all women in the HIPIMR with negative, benign, or probably benign (BI-RADS 1, 2, or 3) BUS visit within one year of a negative screening mammography visit and no personal history of breast cancer. We have clarified this in the Results section. 

Comment 5: This section [statistical analysis] is missing. Evaluation metrics fall under this sub-section. Approach used to determine the ground truth and statistical comparison/significance need to be discussed in this section.

Response 5: Thank you for noting this omission. We have added a statistical analysis section, in which we detail the labeling procedure for included images and note that performance is assessed via computation of sensitivity and specificity. We also define the parameters for dataset selection in this new section. 

Comment 6: The ground truth definition may differ from person to person. In this study, ground truth defined by same two annotators or developers for internal and external dataset. This could explain extraordinarily high performance of a CAD software. This indicates the possible bias in the study design.

Response 6: We appreciate the editor’s comment about possible bias in the study design. Importantly, BUSClean is neither a CADe or CADx software, performing neither detection nor diagnosis of any medical condition. BUSClean identifies, cleans, and extracts information from BUS scan artifacts and abnormalities. The attributes which BUSClean detects are neither subjective criteria nor require specialized training to identify (such as in delineating breast lesions), resulting in little risk of substantive label bias. 

Response to Reviewer 

Comment 1: Some expressions are somewhat subjective and lack objective evidence. It is recommended to cite some references to show that the author's point of view is supported. Especially in the first paragraph of introduction.

Response 1: Thank you for this comment. To strengthen the support for our point of view that medical data cleaning is important for AI development and robustness, we have added references to the following works in the Introduction:

● Șerbănescu M-S, Rotaru-Zăvăleanu A-D, Istrate-Ofițeru A-M, Maria BE-I-A, Enache I-A, Nagy RD, et al., editors. Medical Image Data Cleansing for Machine Learning: A Must in the Evidence-Based Medicine? International Conference on Advancements of Medicine and Health Care through Technology; 2022: Springer.

● Guo M, Wang Y, Yang Q, Li R, Zhao Y, Li C, et al. Normal Workflow and Key Strategies for Data Cleaning Toward Real-World Data. Interactive Journal of Medical Research. 2023;12(1):e44310.

● Willemink MJ, Koszek WA, Hardell C, Wu J, Fleischmann D, Harvey H, et al. Preparing medical imaging data for machine learning. Radiology. 2020;295(1):4-15.

Comment 2: The importance of the research topic is not sufficient. The author points out that various defects in BUS mislead deep learning models, and gives examples of pre-processing in BUSI systems developed by previous studies, but it is not enough to explain what happens without pre-processing.

Response 2: Thank you for this comment. We have identified several studies which note the Clever Hans effect being present in medical AI systems which may be attributed to sub-standard preprocessing to highlight the important of preprocessing like BUSClean enables. See the following added references to these works into the Methods section. 

● Wallis D, Buvat I. Clever Hans effect found in a widely used brain tumour MRI dataset. Medical image analysis. 2022;77:102368.

● Kovács DP, McCorkindale W, Lee AA. Quantitative interpretation explains machine learning models for chemical reaction prediction and uncovers bias. Nature communications. 2021;12(1):1695.

● Bottani S, Burgos N, Maire A, Saracino D, Ströer S, Dormont D, et al. Evaluation of MRI-based machine learning approaches for computer-aided diagnosis of dementia in a clinical data warehouse. Medical Image Analysis. 2023;89:102903.

Comment 3: The authors point out that unlike other imaging data, BUS data contains some defects. However, mere description is very unspecific and difficult to understand. What are the flaws in BUS? What can the software solve? Or all of them? It is suggested that the author include some images for each defect to make it easier for the reader to understand the importance of preprocessing.

Response 3: Figure 6 in the manuscript shows examples of all possible defects in BUS images, labeled according to the BUSClean categories. Additionally, to provide more context, we have added an example description of all abnormalities which can be present in BUS scans into the introduction in response to this comment. 

Comment 4: I'm not absolutely sure, but it seems that Table 2 and Table 3 are not confusion matrices, right?

Response 4: Tables 2 and 3 represent confusion matrices, but are presented in a different format than is typical to allow for the inclusion of the many types of artifacts BUSClean identifies. A confusion matrix can be defined as a table that shows the number of ground truth instances of a class against the number of predicted class instances. 

Comment 5: In the performance section, the authors describe the sensitivity and specificity of the method on various identifications. But why not include these results in the table? In addition, why did the author not consider other evaluation indicators such as accuracy, precision, f1 score and AUC for the report? Is such a short result section sufficient to verify the performance superiority of the proposed software?

Response 5: Thank you for this comment. In response, we have added sensitivity, specificity, and F1 score of all attributes to Tables 2 and 3. We have declined to include accuracy as there is risk of bias, as many of the scan attributes represent very imbalanced classification problems. We have declined to include precision in the interest of space. We have also declined to include AUC as it is not well-suited to the problem, as BUSClean’s filtering and identification can only be represented as hard binary classification, rather than resulting in probabilistic predictions which make AUC well-suited. 

Comment 6: The author's verification of the software developed is not sufficient. The authors only report the results of the software's handling of various defects. However, as this software is a BUS preprocessing software, the author should investigate other evaluation methods for medical image preprocessing. Should authors consider finding out the latest centralized BUS image preprocessing methods and comparing them to show the performance superiority of the software? For example: Article: "KRC-APM: Key region cutting and artificial prior model for breast cancer recognition in ultrasound images" also proposed a BUS preprocessing method.

Response 6: Thank you for this comment. The referenced KRC-APM method is not a task-agnostic preprocessing method, but crops the BUS image to the area most likely to contain cancer. This method is tuned for a single deep learning task (diagnosis of breast cancer in lesions) and not as widely applicable as BUSClean. Furthermore, we do not limit the women included in the BUSClean test set to only those with cancer or those with lesions, which would make evaluation of KRC-APM method invalid on this dataset. There are no existing open-source BUS preprocessing methods which we could compare BUSClean to for any of the identified tasks. 

Comment 7: The authors point out that flaws in BUS data can mislead AI models. However, there is no proof in the author's experiment that the resulting images processed by the software can improve the performance of the AI model. Should the authors consider conducting ablation experiments to show that the various modules of the software can reasonably handle different defects, and to verify that the sub-process results and final results of the processing can improve the AI model?

Response 7: Thank you for this comment. Image artifacts having the potential to mislead AI models and lead to either artificially high/low results or spurious correlations is a well-documented, general result in the AI literature for both medical and non-medical applications. To support the potential effect of BUSClean, we have added citations from the literature (please see responses to reviewer comments 1 and 2). 

Comment 8: The author introduces the processing principle of each module of the software in sufficient detail. However, it is not clear how the different processes of this software deal with different defects. It is suggested that the author give some diagrams for the intermediate and final results of each processing subprocess, so that readers can be more clear about the effectiveness and importance of preprocessing.

Response 8: Figures 2, 4, 5, and 7 already show intermediate steps and/or results for BUSClean’s enhanced scan mode detection, lesion caliper detection, scan area cropping, and lesion caliper detection (failure cases). There are also additional intermediate results shown in the sample notebooks (SampleArtifacts.ipynb, SampleCropping.ipynb, and SampleTextDetection.ipynb) in the GitHub repository for BUSClean https://github.com/hawaii-ai/bus-cleaning. We anticipate that readers concerned with the minutiae of implementation will seek out the source code. 

Comment 9: The author must be more patient and careful to prepare the drawings in the paper. The few figures that do exist look very blurry. In Figure 1, some arrows are covered by the module, while others are in front of the module. The jagged edges of the modules are also very severe. In Figure 3, why is there an extra black rectangle in the bottom half? Do they have any special meaning? The text in Figure 5 is barely legible. The captions in Figure 6 appear to be heavily double shadowing, making them difficult to recognize. The author needs to improve the quality of the graph and suggests using vector graph.

Response 9: We have checked and confirmed that the figures we are uploading are high-quality vector graphics. Could it be that the reviewer is looking at the low-resolution proofs generated by PLOSOne. To view the full-resolution figures, please click the “Click here to access/download” link on the top right of every figure page. 

Comment 10: The authors' references appear to be slightly out of date and do not indicate the cutting-edge nature of the study. Authors are advised to read and refer to recent literature:

● KRC-APM: Key region cutting and artificial prior model for breast cancer recognition in ultrasound images

● Breast cancer diagnosis using optimized deep convolutional neural network based on transfer learning technique and improved Coati optimization algorithm

● One-step abductive multi-target learning with diverse noisy samples and its application to tumour segmentation for breast cancer

● HoVer-Trans: Anatomy-Aware HoVer-Transformer for ROI-Free Breast Cancer Diagnosis in Ultrasound Images

● REAF: ROI Extraction and Adaptive Fusion for Breast Cancer Diagnosis in Ultrasound Images

Response 10: Thank you for this comment. We have addressed the suitability of each of the requested references below, in the order presented by the reviewer. 

● The KRC-APM method is not appropriate as it cuts to areas with high confidence of containing cancer, which is related to a specific downstream AI task. We crop to the scan area task-agnostically. This task is fundamentally different from the BUSClean offering, limiting applicability to our method. 

● This technique is developed for breast thermography imaging, not breast ultrasound imaging and thus does not provide context relevant to BUSClean. 

● HoVer-Trans does not develop a preprocessing technique, instead presenting an AI diagnosis solution for breast cancer in BUS images, a downstream task to BUSClean. 

● REAF does not develop a preprocessing technique, instead presenting an AI diagnosis solution for breast cancer in BUS images, a downstream task to BUSClean.

---

## [Editor Report · Decision Letter 1]

26 Nov 2024

BUSClean: Open-source software for breast ultrasound image pre-processing and knowledge extraction for medical AI

PONE-D-24-32854R1

Dear Dr. Bunnell,

We’re pleased to inform you that your manuscript has been judged scientifically suitable for publication and will be formally accepted for publication once it meets all outstanding technical requirements.

Kind regards,

Arka Bhowmik, Ph.D.

Academic Editor

PLOS ONE

Additional Editor Comments (optional):

Authors have addressed the comments of the reviewer, and my decision is favorable for the paper.
---

## [Editor Report · Acceptance letter]

2 Dec 2024

PONE-D-24-32854R1 

PLOS ONE

Dear Dr. Bunnell, 

I'm pleased to inform you that your manuscript has been deemed suitable for publication in PLOS ONE. Congratulations! Your manuscript is now being handed over to our production team.

Kind regards, 

on behalf of

Dr. Arka Bhowmik 

Academic Editor

PLOS ONE